# Evaluation of the Intestinal Permeability of Rosmarinic Acid from *Thunbergia laurifolia* Leaf Water Extract in a Caco-2 Cell Model

**DOI:** 10.3390/molecules27123884

**Published:** 2022-06-17

**Authors:** Nanthakarn Woottisin, Sophida Sukprasert, Thitianan Kulsirirat, Thipaporn Tharavanij, Korbtham Sathirakul

**Affiliations:** 1Graduate Program in Integrative Medicine, Chulabhorn International College of Medicine, Thammasart University (Rangsit Campus), Pathum Thani 12120, Thailand; nanthakarn729@gmail.com; 2Division of Integrative Medicine, Chulabhorn International College of Medicine, Thammasart University (Rangsit Campus), Pathum Thani 12120, Thailand; 3Department of Pharmacy, Faculty of Pharmacy, Mahidol University, Bangkok 10400, Thailand; lordrx16@gmail.com; 4Endocrinology and Metabolism Unit, Department of Internal Medicine, Faculty of Medicine, Thammasat University (Rangsit Campus), Pathum Thani 12120, Thailand; thipaporn_t@yahoo.com; 5Center of Excellence in Applied Epidemiology, Thammasat University (Rangsit Campus), Pathum Thani 12120, Thailand

**Keywords:** Caco-2, intestinal absorption, paracellular diffusion, rosmarinic acid, *Thunbergia laurifolia*

## Abstract

*Thunbergia laurifolia* (TL) has been traditionally used as an antidote and an antipyretic drug by folk healers for centuries in Thailand. Rosmarinic acid (RA) is major compound in TL extract and has attracted great interest due to its potential broad pharmacological effects. Herein, the permeability of RA was investigated in TL extract and as a pure compound in a Caco-2 cell model by using high-performance liquid chromatography with a photodiode array detector (HPLC-PDA). The results reveal that the apparent permeability coefficient (*P*_app_) values of RA in TL extracts and pure RA significantly increased after deconjugation by β-glucuronidase/sulfatase enzymes. Our findings exhibit possible saturable biotransformation of RA and/or membrane transport while penetrated through Caco-2 cells. The cumulative amounts of RA as pure compounds and in TL extracts increased with the exposure time, and the efflux ratio (ER) was 0.27–1.14. RA in the TL extract has a similar absorption in the conjugated form and in the pure compound. The intestinal absorption of them is through passive diffusion. Therefore, our findings conclude that the intestinal transport of RA in TL extracts was mainly penetrated as conjugated forms with glucuronic acid and/or sulfate across Caco-2 cells and transported via passive diffusion.

## 1. Introduction

*Thunbergia laurifolia* (TL), commonly known as “Rang Jeud”, has been widely used in traditional Thai medicine for many centuries. It is a popular ornamental plant and a fast-growing vine in the tropics [1]. In Thailand, TL was selected by the Ministry of Public Health to be included in “*National List of Essential Herbal Drugs A.D. 2011*”, as an herbal drug for the treatment of poison symptoms and fever [2]. TL leaf extract is mainly used as an antidote for treating intoxications with insecticides, herbicides, lead, alcohol, and chemical toxins [3,4,5,6]. This plant has been reported to have antioxidant, anti-inflammatory, hepatoprotective, antitumor, and antihyperglycemic activities [7,8,9,10,11]. Previous studies showed that TL leaves contain seven chlorophyll derivatives, one major carotenoid, and lutein from acetone and ethanol extractions [12]; apigenin, caffeic acid, gallic acid, protocatechuic acid, and vitexin from water extraction [13,14]; and rosmarinic acid (RA) from both ethanol and water extractions [15,16].

Our previous study demonstrated that the water leaf extract of TL contained RA as the main component [17]. RA is a polyphenol compound, which is a widely occurring natural product in many plants, such as rosemary, perilla, self-heal, and lemon balm, with potential for drug application in patients with seasonal allergic rhinoconjunctivitis [18]. RA’s structure is a molecule with slightly hydrophilic properties. The solubility of this molecule is higher in most organic solvents [19]. Its water solubility is estimated in silico to be 0.041 g/L (ALOGPS). The molecular weight of RA is 360.3 Da. A logarithm of octanol–water partition coefficient (logP) of RA is 1.626. RA has 5 and 8 H-bond donors and acceptors, respectively [20]. RA has gained increased attention due to its various pharmacological effects, such as anti-inflammatory, antioxidant, antidiabetic, and hepatoprotection [18]. It is characterized by broad pharmacological activities that make it interesting in drug development.

The activity and toxicity of RA rely on its pharmacokinetics. One of the most important factors in determining the success of using medicinal herb extracts is the bioavailability (BA) of the active compounds in an extract. RA is one of the well-known active compounds available in over 160 species of herbal plants. However, toxicity and BA are still issues for investigation in product development [18]. The content of RA varies among the species of the same genus, subspecies of the same species, samples collected in different seasons of the year, etc. [21]. The absolute BA of RA in rats was determined by comparing pharmacokinetic data after the administration of pure RA as a single oral dose (12.5, 25, and 50 mg/kg) and intravenously (0.625 mg/kg). RA was determined to have poor oral absolute BA (1.69%, 1.28%, and 0.91% for 12.5 mg/kg, 25 mg/kg, and 50 mg/kg, respectively). Moreover, a lack of dose proportionality was observed in this study [22]. The pharmacokinetic characteristics of pure RA and RA from natural products in rats were reviewed [21]. Moreover, the pharmacokinetics of RA in a human was also summarized in the current publication [20]. There is no absolute BA of RA in humans. There is a limited number of pharmacokinetic studies of RA in humans from both pure and herbal extracts [20,23]. Interestingly, differences in the metabolism of RA between humans and rats were reported. The differences in the location of enzymes, the affinity of substances, and characteristics of the enzymatic reactions are considered causes of the differences in pharmacokinetics [24].

RA must be bioavailable to the systemic circulation in order to be transported to the target tissues to produce its pharmacological effects. Thus, the systemic BA of RA from an extract is a key issue in determining how fast and to what extent RA can reach systemic circulation [25]. The systemic absorption of the oral dosage form depends on two processes: One is the dissolution, and the other is the permeability. RA has low hydrophilicity and should be noted as the biopharmaceutical classification (BCS) classes III or IV [26]. Due to its low aqueous solubility and low permeability, both aqueous solubility and permeation are the rate-limiting processes for systemic absorption [27,28], and, consequently, the BA of RA is low. To the best of our knowledge, there are no pharmacokinetic studies of RA in this extract regarding the TL extracts. The BA of RA in the TL extracts from the gastrointestinal tract is still being investigated. The aforementioned permeability process is one of the essential processes for systemic absorption; therefore Caco-2 cells, which are derived from human colon adenocarcinoma and which resemble morphologically the enterocytes of the small intestine [29], were selected in this study to investigate the BA of the RA in the extracts. Intestinal transporter proteins and metabolizing enzymes work together and have crucial roles in regulating the oral absorption of various xenobiotics. In Caco-2 cell monolayers, the transporter proteins, efflux proteins, and metabolic phase II conjugation enzymes are available [29,30,31]. Moreover, the Caco-2 cell model can be used to determine the transport mechanism. The bidirectional permeability from the apical (A) to basolateral (B) direction and vice versa can be detected in the Caco-2 cell system. The efflux ratio (ER) can be calculated from the ratio of this bidirectional permeability. Therefore, the Caco-2 cell can be one of the good models for in vitro prediction of intestinal drug permeability and absorption of compounds from medicinal herbal extracts in vitro [32,33]. The Caco-2 cell model was used to evaluate the permeability of polyphenol and phenolic compounds [34,35]. The permeability of pure RA was investigated for the first time in the Caco-2 cell model in 2005 [36]. RA was shown to be transported via paracellular passive diffusion. The monocarboxylic acid transporter (MCT) does not get involved in the permeability of RA. Due to coexisting compounds in the herbal extract, the interaction among coexisting compounds may interfere with the absorption of active compounds [37,38]. Thus, it is crucial to detect the permeability of not only pure RA but also RA in the herbal extract. The permeability of RA was investigated in both pure RA and RA in *Prunella vulgaris* herbal extract [39]. The permeabilities of polyphenols and terpenoids in rosemary (*Rosmarinus officinalis* L.) extract were also investigated in Caco-2 cells. A similar uptake of RA was confirmed in the pure form and herbal extract form. Moreover, RA was transferred across Caco-2 cells almost entirely in the conjugated form. There was no plant matrix effect on the efficacy of the RA being predicted in this study [26]. In contrast, the study of BA and the metabolism of rosemary infusion polyphenols using the Caco-2 and HepG2 cell model systems showed that RA in the rosemary infusion showed improved BA compared to pure RA [40]. In addition, the study of the effect of flavonoids, luteolin, and apigenin on the BA of RA in the decoction extract of *Plectranthus barbatus* leaves was observed in the Caco-2 cell model. The BA of RA in the herbal extract showed higher BA than that in the pure compound. This was the first finding demonstrating that RA seems to be a substrate of MCT and organic anion transporter (OATP) but not the substrate of p-glycoprotein (p-gp). This result was inconsistent with the result from pure RA as mentioned above [41].

At present, there is no evidence of the permeability and pharmacokinetics of the TL leaf extract even though it has been traditionally prepared and prescribed in the form of a capsule or decocted as a herbal tea for treatment via oral administration according to the Thai National List of Essential Medicines (NLEM) [2]. Therefore, the absorption mechanism of RA in this extract via Caco-2 cell monolayers should be investigated to determine the BA of this compound in the extract. Therefore, the aim of the present study was to determine the permeability of RA in the TL leaf extract in the Caco-2 cell model for ethnological use in the future.

## 2. Results

### 2.1. Determination of RA in TL Water Extract

Under high-performance liquid chromatography (HPLC) conditions, the retention time of RA in the TL extract and pure compound was 19.2 min (Figure 1). A calibration curve of RA was obtained over a concentration range of 0.025–1 µg/mL to quantify the concentration of RA in the samples. The linear regression of RA was *y* = 105197*x* − 111.48 with a correlation coefficient (r^2^) = 0.9999, which meant the back-calculated concentrations of the calibration standards were within ±15% of the nominal value. All these indicate good linearity in this calibration curve and that the equation could be used to determine the concentration of RA in this work. The amount of RA in the TL water extract was 13.07 mg/g extract.

The lower limit of detection (LLOD) of RA was the analyzed signal at a concentration of 0.025 µg/mL, which was at least 5 times the signal of a blank sample. The mean concentration of RA across 0.025–0.8 µg/mL within and between-run accuracy and precision values was not exceeded by 15% for the QC samples and did not exceeded 20% for the lower limit of quantification (LLOQ) recommended by the European Medicines Agency (EMA) for bioanalytical method validation [42].

### 2.2. Stability of the Samples

The percentage remaining for short-term (stored at room temperature for 24 h) and long-term (stored at −80 °C for 30 days) stability tests showed that RA in the extract was highly stable, up to 97% in both conditions (Table 1).

The stability of RA in the TL extract of 250 µg/mL, which was equivalent to pure RA of 3.267 µg/mL in Hank’s balanced salts solution (HBSS) is shown in Figure 2. The RA stability was 99% and 91%, respectively, from the extract and pure. No statistically significant difference in RA in the extract between before and after incubation was observed (*p* > 0.05). However, the stability of pure RA was significantly decreased after incubating for 2 h.

### 2.3. Cytotoxicity Test

As shown in Figure 3, the TL extract and RA were nontoxic to Caco-2 cells at concentrations below 500 µg/mL. However, the cell viability of the TL extract and RA at 1000 µg/mL was 74% and 85%, respectively. Therefore, the proposed working noncytotoxic concentrations for the TL extract in concentrations of 250 and 500 µg/mL were used in the cell monolayer transport experiment.

### 2.4. The Impact of Post Permeability Deconjugation Treatment with β-Glucuronidase/Sulfatase on Apparent Permeability Coefficient (P_app_) of RA from TL Extract and of Pure Compound

The TEER value of the Caco-2 model was measured to ensure that the Caco-2 cell monolayers were confluent and suitable for the permeability study. When the culture time was increased, TEER values gradually increased over time and reached 400 Ω·cm^2^ in 21 days.

#### 2.4.1. In Vitro Permeability Study

The permeability before deconjugation of RA in the TL extract with different concentrations is shown in Table 2 and Figure 4A. The *P*_app_ values of RA from the TL extract in a concentration of 250 µg/mL in both the A to B (*P*_app_ (A-B)) and B to A directions (*P*_app_ (B-A )) were 4.68 ± 0.22 × 10^−6^ cm/s and 6.13 ± 0.36 × 10^−6^ cm/s, respectively, while in the TL extract in a concentration of 500 µg/mL, the values were 5.33 ± 0.25 × 10^−6^ cm/s and 6.09 ± 0.05 × 10^−6^ cm/s, respectively. The *P*_app_ (B-A ) values for RA in both the 250 µg/mL and 500 µg/mL of TL extracts were significantly different (*p* < 0.01) from the *P*_app_ (A-B) values for the same doses. However, only the *P*_app_ (A-B) value of RA in the 250 µg/mL of TL extract was significantly different from that of RA in the 500 µg/mL of TL extract (*p* < 0.05). In contrast, all *P*_app_ values from pure RA could not be detected in this study.

After treatment with β-glucuronidase/sulfatase enzymes, the *P*_app_ (A-B) and (B-A ) values of RA in the 250 µg/mL TL extract were 46.05 ± 1.15 × 10^−6^ cm/s and 19.90 ± 0.34 × 10^−6^ cm/s, respectively, while those values in the 500 µg/mL TL extract were 27.27 ± 3.32 × 10^−6^ cm/s and 12.74 ± 0.20 × 10^−6^ cm/s, respectively (Table 2 and Figure 4B).

The permeability of RA in a pure compound with different concentrations could be detected only after deconjugation of the permeate in the acceptor compartment from transport experiment (Table 2 and Figure 4B). The *P*_app_ values of RA as a pure compound in a concentration of 3.267 µg/mL were 41.19 ± 2.29 × 10^−6^ cm/s and 20.35 ± 5.16 × 10^−6^ cm/s for both the A-B and B-A directions, respectively, while the *P*_app_ values of RA as a pure compound in a concentration of 6.808 µg/mL were 22.88 ± 0.97 × 10^−6^ cm/s and 6.24 ± 2 0.23 × 10^−6^ cm/s for both the A-B and B-A directions. The *P*_app_ (A-B) values of pure RA in concentrations of 3.267 and 6.808 µg/mL were 2.02- and 3.67-fold greater than those in the B-A direction, respectively.

#### 2.4.2. Cumulative Amounts

The cumulative permeations of RA before deconjugation for both the low (250 µg/mL) and high (500 µg/mL) doses of the TL extracts were markedly increased and did not reach the plateau within 120 min in both the A-B and B-A directions (Figure 5A,B). The cumulative permeation of RA in the high dose of TL was proportional to that of RA in the lower dose. However, the pure RA could not be detected within 120 min in both the A-B and B-A directions. After deconjugation of the permeates, significant increases in RA were detected in all experiments. Likewise, RA in the extract and pure RA also tended to increase and become unsaturated after deconjugation (Figure 5C,D).

#### 2.4.3. Basolateral Recovery

Basolateral recoveries of RA in the 250 and 500 µg/mL TL extracts significantly increased from 1.3 ± 0.1% to 13.5 ± 0.8 (*p* < 0.001) and from 1.3 ± 0.1% to 7.5 ± 0.6% (*p* < 0.01). Deconjugation resulted in about 10 and 5 times during the 2 h uptake study compared to before deconjugation, respectively (Figure 6).

#### 2.4.4. ER Values

The ER values of RA in the TL extracts were 1.3 and 1.1 before deconjugation and 0.4 and 0.5 after deconjugation in the concentrations of 250 and 500 µg/mL, respectively. Moreover, the ER values of RA as a pure compound in 3.267 and 6.808 µg/mL were 0.5 and 0.3, respectively (Table 2).

## 3. Discussion

In this study, we demonstrated for the first time the intestinal permeability of RA in TL leaf water extracts and pure RA by using an in vitro Caco-2 cell model. The *P*_app_ values of RA were different before deconjugation by β-glucuronidase/sulfatase enzymes. The permeability of RA was not detectable in both the A-B and B-A directions for pure RA. However, after deconjugation, RA was detectable in all samples. The increases in RA after deconjugation were to different extents. For RA in TL leaf water extract, the *P*_app_ values increased more in the 250 µg/mL than those in the 500 µg/mL in both the A-B and B-A directions. RA in the TL leaf water extracts and pure RA demonstrated passive transport with no active efflux (ER was less than 2). Thus, as shown in Figure 5A,B, the cumulative amounts of RA (before deconjugation) were proportionally changed when the dose increased from 250 µg/mL to 500 µg/mL. In contrast, as shown in Figure 5C,D, the cumulative amounts of RA (after deconjugation) were not proportionally changed. These results show the saturation of the process related to the conjugation of RA. These saturable processes can be the conjugation process itself and/or the membrane transport of the conjugated RA. The glucuronide and sulfate metabolites of drugs typically have limited cell membrane permeability, and, subsequently, their distribution and membrane transport require transport proteins [43,44]. These processes can be saturable.

RA was found to be the most abundant in the TL water extract, which is consistent with previous studies [14,15,16]; therefore, it was quantified and used as a trace compound in this study even though the *P*_app_ values of RA in the TL extracts and pure RA increased after treatment with the enzymes in all samples, and these findings have confirmed that phase II conjugation occurred during the penetration of RA through Caco-2 cells. However, the increment of *P*_app_ values of RA seems to be higher in the 250 µg/mL dose as illustrated in Table 2. In the A-B direction, the *P*_app_ value of RA in the 250 µg/mL dose increased almost 10-fold (4.7 ± 0.2 × 10^−6^ cm/s to 46.1 ± 1.22 × 10^−6^ cm/s), whereas that of the *P*_app_ value of RA in the 500 µg/mL dose increased only 5–6-fold (5.3 ± 0.3 × 10^−6^ cm/s to 27.3 ± 3.3 × 10^−6^ cm/s). The increments in the *P*_app_ values of RA in the B-A direction were less sensitive to deconjugation. The *P*_app_ value of RA in the 250 µg/mL dose in the B-A direction increased 3-fold (6.1 ± 0.4 × 10^−6^ cm/s to 19.9 ± 0.3 × 10^−6^ cm/s), whereas that of the *P*_app_ value of RA in the 500 µg/mL dose only doubled (6.1 ± 0.1 × 10^−6^ cm/s to 12.7 ± 0.2 × 10^−6^ cm/s). Thus, the extent of the conjugation of RA depends on the direction and dose.

For RA in the TL leaf water extract, after deconjugation, the *P*_app_ values increased more in 250 µg/mL than those in 500 µg/mL in both the A-B and B-A directions. This finding was consistent with previous clinical studies reported that the conjugated RA could be detected in human blood samples after the administration of RA containing *Melissa officinalis* extract. Moreover, comparing the two doses (250 and 500 mg of RA) in the same study, it was evident that an increase in dosage was followed by an increase in free RA concentration, while the concentration of conjugated forms increased insignificantly [45]. This suggests that conjugational processes are characterized by saturation kinetics [20]. Moreover, the permeability of RA in the TL extracts was significantly higher in both the A-B and B-A directions. As these results show, some coexisting compounds in the TL extract might affect the permeability of RA by unidentified mechanisms. The interaction and coexistence of the compounds in the natural extract may be important for the therapy by the herbal extract [32,33,34,46]. This interaction can be synergistic or antagonistic to pharmacokinetics or pharmacodynamics of the active compounds in the extract. For example, the co-existence of luteolin and apigenin in the decoction extract of *Plectranthus barbatus* leaves can improve the BA of RA, which is an active compound in the extract [41]. In addition, the rosemary infusion matrix enhanced the BA of RA compared to that of the pure compound [40]. The varying types and amounts of components inside each medicinal herb extract may interfere with the BA of RA for ethnological use. However, to identify the mechanism, further transport studies with inhibitors are essential. These studies are under investigation in our laboratory.

In addition, other intestinal Phase II enzymes (e.g., UDP-glucuronosyltransferases (UGTs), sulfotransferases, and glutathione S-transferases (GSTs)) might be responsible for enhancing or preventing intestinal absorption [29]. Therefore, it is possible that RA in the TL extract was transported across Caco-2 cells and extensively conjugated with sulfate and glucuronide during the absorption process, which was consistent with the absorption of RA in pure RA and other plant extracts as previously reported in [39]. Based on our results, the *P*_app_ of RA could not be detected when pure RA was used, while the *P*_app_ of RA could be detected in TL extracts that contained equivalent RA amounts. Thus, it is possible that the co-existence of the compounds in the TL extract may enhance the permeability of RA. It would be assumed that RA in the TL extract was mainly transferred from the intestine into the bloodstream in the form of the conjugation of parent compounds (glucuronidation or sulfation), which are presumed to decrease bioactivity because of their changed structures. However, the potential bioactivity in vivo depends on the absorption, distribution, metabolism, and excretion of the compound within the body after ingestion [47].

Physicochemical factors, such as lipophilicity and solubility, are the key properties of permeability [48]. Most polyphenols are probably too hydrophilic to penetrate the enterocytes by passive diffusion via transcellular transport [49]. The ionized and hydrophilic molecules can penetrate via paracellular transport in enterocyte. The proportion of unionized and ionized species depends on the pH and pKa of the molecule. The ionized species are less lipophilic and are less able to pass through a lipid bilayer of the CaCo-2 cell. The less lipophilic species can passively diffuse via the paracellular route. The unionized species can penetrate via the transcellular transport and can be transformed into glucuronidase or sulfated forms by the phase II xenobiotic metabolizing enzymes [26]. RA was reported to be transported mainly via the paracellular route [30]. However, due to the extensive detection of the conjugated form in this study after permeation in the Caco-2 cell model, the transcellular transport persisted. Due to its poor aqueous solubility, low BA and low permeability through biological barriers cause limitations to its therapeutic applications. RA showed poor water solubility, low partition coefficient (log Kow = 1.82) [50], and low permeability; 0.03–0.06% of RA was absorbed in Caco-2 cells [36]. RA can be classified in the BCS classes III or IV [26,51]. Thus, RA has low BA in vivo [20,22,23].

RA might lead to incomplete or low absorption as mentioned in previous studies [22,36]. Our investigation showed that RA in the TL extract and pure RA had low permeability, but also intensive conjugation in Caco-2 cells. Their *P*_app_ increased after deconjugation by phase II xenobiotic metabolizing enzymes in the intestine, resulting in good permeability. Additionally, after the administration of *Melissa officinalis* extract containing 100 mg RA, RA could not be detected by the High-Performance Liquid Chromatography-Electrochemical Detector (HPLC-ECD) in almost all of the human volunteers in this clinical study [45]. However, the RA could be detected after deconjugation. This study is consistent with our ongoing clinical study (unpublished data). Although we used a more sensitive bioanalytical method, LC-MS/MS with an LLOQ and LLOD of 5 and 0.75 ng/mL, respectively, RA could not be detected in plasma samples for almost all of the volunteers after the administration of the TL extract.

The cumulative amounts of RA in the TL extracts and pure RA indicated that they were absorbed via passive transport, which was consistent with previous reports [36,39]. The bidirectional intestinal transport of RA in TL extracts and pure RA was investigated to determine whether there was an active efflux. Our findings show that the transport of RA in TL extracts and pure RA across Caco-2 cell monolayers was less than 2, which could be inferred that they were not involved in active efflux [52]. Therefore, it was possible that RA might be paracellularly absorbed [36] and conjugated with glucuronides and/or sulfate.

## 4. Materials and Methods

### 4.1. Plant Extraction

Mature leaves of TL were harvested from a cultivated area in Yasothon Province, Thailand (from June to July). The botanical identification was authenticated by the Bangkok Herbarium Office, Department of Agriculture, Ministry of Agriculture and Cooperatives, Bangkok, Thailand (BK. No. 069396). The process of herbal extraction in this study imitated the ethnological usage in the form of TL infusion and crude powder. Leaves were air dried at 40–50 °C for 5 days, ground to a coarse powder, and boiled at 100 °C for 15 min. They were filtered and dried by lyophilization. Then, the extract was stored frozen at −20 °C until use.

### 4.2. Chemicals and Reagents

RA (purified ≥ 98%) was purchased from Sigma-Aldrich (St. Louis, MO, USA). Water used in this study was purified using the Milli-Q water purification system (WATER PRO^®^ PS, MO, USA). All other chemicals were of analytical grade and purchased from local chemical suppliers. HPLC-grade solvents obtained from Fisher Scientific (Seoul, Korea) were used for drug analysis. Dulbecco’s modified eagle’s medium (DMEM), HBSS Trypsin-EDTA solution, and Penicillin and Streptomycin solution were purchased from Gibco^TM^ (Life Technologies, Grand Island, NY, USA). Sodium pyruvate solution was purchased from Calbiochem^®^ (Merck, Germany). Fetal bovine serum (FBS) was purchased from Hyclone^®^ (GE Healthcare Life Sciences, Salt Lake, UT, USA). Twenty-four well cell culture plates were purchased from Costar (Corning Incorporated, Corning, NY, USA). Millicell^®^ Cell Culture inserts and Millipore Express^®^ PES Membrane (0.22 μm) were purchased from MILLIPORE (USA).

### 4.3. Experimental Procedures

#### 4.3.1. HPLC Analysis

TL extracts and RA were dissolved in HBSS to be used as samples and the standard, respectively. HPLC analysis was performed on a Shimadzu LC-20A (Japan) equipped with photodiode array detector (PDA). A reverse-phase analytical C_18_ column (250 mm × 4.6 mm × 5 µm; ACE^®^ Hichrom, UK) was used at room temperature. The mobile phase was 1% formic acid in water (solvent A) and acetonitrile (solvent B) using gradient elution. Initial solvent of 15% to 40% in B for 20 min was used, followed by a linear gradient to 100% in B for 10 min and back to 15% in B at 35 min and a re-equilibrated column for 5 min prior the initial condition. The flow rate was set at 1.0 mL/min at 25 °C. The detector was set at the wavelength of 280 nm, and the injection volume was 50 µL.

#### 4.3.2. Stability Study

TL water extract was stored at two different conditions, consisting of room temperature for 24 h (short-term stability) and −80 °C for 30 days (long-term stability). The samples were calculated to have a percentage remaining using the mean values of RA concentrations in TL water extract before and after stability tests.

The stability of RA in the TL extract of 250 µg/mL, which was equivalent to pure RA of 3.267 µg/mL in HBSS solution was determined. The TL extracts and RA were dissolved in HBSS solution (pH 7.4) and incubated at 37 °C for 2 h according to the same conditions as in the transport experiment. The contents of the RA as in TL extract and pure compound were measured for stability tests using HPLC. Their RA remains before and after incubation were calculated and expressed as a percentage of RA.

#### 4.3.3. Transepithelial Transport Experiment

Caco-2 cells (ATCC no. HTB-37) were obtained from the American Type Culture Collection (Rockville, MD, USA) and cultured in DMEM containing 100 mM sodium pyruvate, 20% fetal bovine serum, and 10,000 U/mL penicillin and 10,000 ug/mL streptomycin. Cell culture was maintained at 37 °C in a humidified atmosphere of 5% CO_2_. On achieving 80–90% confluence, Caco-2 cells were harvested using 0.25% trypsin–EDTA solution and mixed with complete DMEM medium. All cells were used for transport experiments between passages 30 and 40.

Cell viability of TL extracts and RA on Caco-2 cells was determined using MTT following modified assay [53]. Caco-2 cells were seeded in 96-well plates at a density of 2 × 10^4^ cells/well and cultured at 37 °C and 5% CO_2_ for 24 h. TL extracts and RA were dissolved in 50% DMSO and diluted with complete DMEM medium containing 0.1% DMSO in the culture medium. The tested compounds were diluted with medium at a concentration range of 1–1000 µg/mL for 24 h exposure experiment.

After removal of the medium, the cells were washed with PBS and incubated with 50 µL of 0.5% MTT solution for 2 h at 37 °C and 5% CO_2_. A total of 50 μL of DMSO was put into each well to dissolve the formazan crystals. The absorbance was detected using a microplate reader at 590 nm. The percentage of cell viability was calculated according to the following Equation (1):(1)$$\%\;\mathrm{Cell}\;\mathrm{viability}=\frac{A\;\left(\mathrm{sample}\right)-A\;\left(\mathrm{blank}\right)}{A\;\left(\mathrm{control}\right)-A\;\left(\mathrm{blank}\right)}\times 100$$
where A (sample) is absorbance value of sample; A (blank) is absorbance value of blank, and A (negative control) is absorbance value of control (cells).

The experiment was performed in three replicates, and the concentration with a viability of more than 90% was chosen for evaluation in monolayer transport experiments as nontoxic to cells.

The integrity of the Caco-2 cell monolayer in the transwell plates was evaluated by measuring transepithelial electrical resistance (TEER) across the monolayers using Millicell-ERS VoltOhm Meter (Millipore, Temecula, CA, USA). TEER was measured at eight time points (i.e., on days 1, 3, 5, 7, 10, 14, 20, and 21) and at two points on the day before and after the transport experiment to reflect the tightness of intercellular junctions. The cells with a TEER ≥ 400 Ω·cm^2^ were used for this experiment. The TEER values of Caco-2 cell monolayers were calculated by the following Equation (2):(2)$$\mathrm{TEER}\;\left(\Omega \cdot {\mathrm{cm}}^{2}\right)=\left[\mathrm{TEER}\;\left(\Omega \right){\mathrm{TEER}}_{\mathrm{background}}\;\left(\Omega \right)\right]\times A\;\left({\mathrm{cm}}^{2}\right)$$
where TEER (Ω) is the transepithelial electrical resistance across Caco-2 cell monolayers, and TEER_background_ (Ω) is that across the insert only (without cells). A (cm^2^) is the area of the insert, 1.12 cm^2^. The TEER value was not less than 400 Ω·cm^2^ after the experiment.

Caco-2 cells were seeded at a density of 3.2–3.6 × 10^4^ cells/cm^2^ on a 0.4 µM polycarbonate membrane, a 12 mm insert (Corning Incorporated, Corning, NY, USA) in 24-transwell permeable supports and grown in a humidified chamber (37 °C, 5% CO_2_). The culture medium in inserts and plates was changed every 2–3 days for both A and B sides. All cells were used for transport experiments on days 19–21 postseeding.

Before starting the transport experiment, the inserts were washed twice and equilibrated with transport buffer (HBSS containing 1 M HEPES, pH 7.4) for 30 min. Sample solutions of TL extract and RA were prepared and diluted in HBSS to the desired final concentrations. For the transport study of TL extract and RA across Caco-2 cell, sample solutions were added on donor chamber, 400 µL on A side or 1200 µL on B side. The equal volumes of transport buffer were replaced in the receiving chambers. The RA transport in samples was assessed either in A-B or B-A direction across Caco-2 cell monolayers. The plates were incubated at 37 °C for 120 min with gentle shaking at 100 rpm. The solutions of 200 µL on A side and 600 µL on B side were collected at the selected times of 30, 60, 90, and 120 min and immediately replaced with an equal volume of transport buffer to maintain the total volume in receiving chamber. Samples were stored at −80°C until analysis. The amount of transported RA in samples and reference compound was calculated using HPLC system according to the standard curve.

#### 4.3.4. Transepithelial Transfer of RA in TL Extract and Pure RA after Treating with β-Glucuronidase/Sulfatase

β-glucuronidase/sulfatase (Type H-2 from Helix pomatia, 85 units/L of glucuronidase, and 7.5 units/L of sulfatase, Sigma-Aldrich Co., St. Louis, MO, USA) was added to collected samples, which were obtained from B-A solutions at 30, 60, 90, and 120 min, in a ratio of 1:20 and incubated at 37 °C for 4 h to release the parent compounds. These samples from both sides were then injected directly and analyzed for RA by HPLC.

The relationship of Q (cumulative permeated amount; μg) versus time, was calculated with the following Equation (3):(3)$$Q=\mathrm{CnV}+\overset{n-1}{\underset{i=1}{\sum }}\mathrm{CiS}$$
where Cn is concentration of RA determined at nth sampling interval; V is volume of individual side, $\overset{n-1}{\underset{i=1}{\sum }}\mathrm{Ci}$ is sum of concentration of RA determined at sampling intervals 1 through n − 1, and S is volume of sampling aliquot

The *P*_app_ (unit: cm/s) was determined and calculated from the amount of compound permeated per time by the following Equation (4):(4)$$P_{\mathrm{app}}=\left(\frac{dQ/dt}{C_{0}\times A}\right)$$
where *dQ*/*dt* is the steady-state flux of test compound across the cell monolayer in the receiver chamber (µg/min); *C*_0_ is the initial concentration in the donor (A or B side) chamber (µg/mL), and *A* is the surface area of the cell monolayer (cm^2^). The membrane area (cm^2^) of the insert is 1.12 cm^2^.

The ER was determined by calculating the ratio of *P*_app_ (B-A) versus *P*_app_ (A-B) as the following Equation (5):(5)$$\mathrm{ER}=\frac{P_{\mathrm{app}}\;\left(\mathrm{B-A}\right)}{P_{\mathrm{app}}\;\left(\mathrm{A-B}\right)}$$
where *P*_app_ (B-A) is the apparent permeability coefficient value in B-A direction, and *P*_app_ (A-B) is the permeability coefficient value in A-B direction.

Basolateral recovery (%) was calculated as the amount of test compound transported through the Caco-2 cell monolayer to the B side and remaining on the A side at the end of the experiment divided by the initial amount on the A side.

### 4.4. Statistical Analysis

Data were presented as mean ± S.D. Statistical significance was analyzed by Student’s *t*-test for the difference between groups and pair-*t* test for the difference within groups. Differences at a value of *p* < 0.05, *p* < 0.01, and *p* < 0.001 were considered statistically significant. SPSS was used for all statistical analyses.

## 5. Conclusions

This is the first investigation in vitro into the intestinal absorption of RA as the main compound in TL leaf water extracts compared with pure RA in two different equivalent doses using the Caco-2 cell model. RA penetrated through Caco-2 cell monolayers and was detected by HPLC-PDA. RA in TL leaf water extracts could be detected in both A-B and B-A directions unlike pure RA. Thus, this study is evidence of the extensive biotransformation of RA while penetrating through Caco-2 cells. It is feasible that the coexistence of the compounds in the TL extract may enhance the permeability of RA in this extract. The intestinal absorption of RA is mainly via passive paracellular transport and might not involve active efflux transport. Moreover, after deconjugation of penetrates from the Caco-2-cell model by using β-glucuronidase/sulfatase enzymes, total RA could be detected in all samples. Due to dose-disproportionate increases in total RA after deconjugation, the saturable biotransformation and/or membrane transport of conjugated RA may exist. However, further studies must be carried out with enzyme inhibitors to elucidate this mechanism.

## Figures and Tables

**Figure 1 molecules-27-03884-f001:**
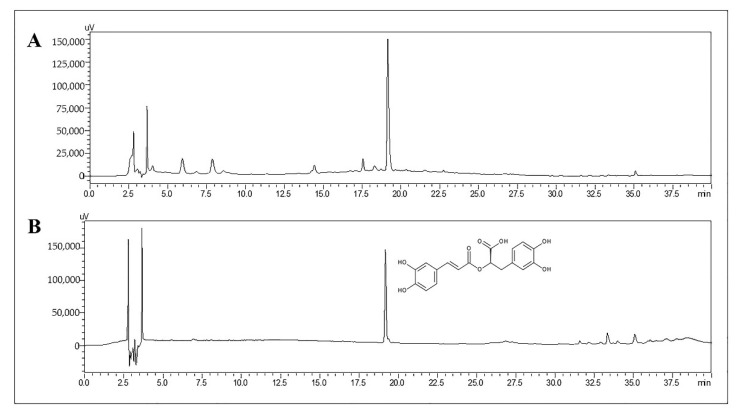
Representative HPLC chromatogram of TL extract (**A**) and RA (**B**).

**Figure 2 molecules-27-03884-f002:**
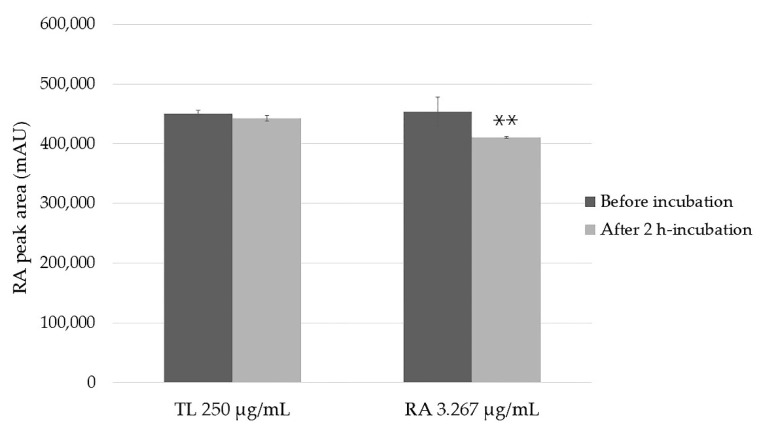
The stability of RA in TL extract and pure compound in HBSS (pH 7.4) before and after incubation at 37 °C for 2 h. The data represent the mean ± SD from three replicates. ** was significantly different from RA in TL extract at 250 µg/mL (*p* < 0.01).

**Figure 3 molecules-27-03884-f003:**
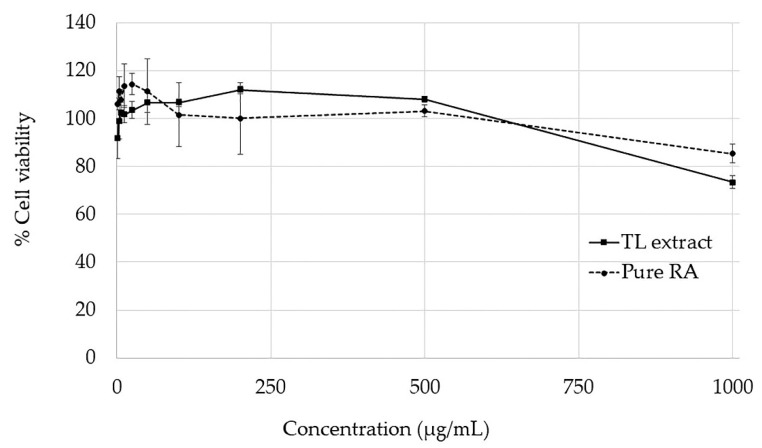
Cytotoxicity of the TL water extract and pure RA was investigated on Caco-2 cells after incubation at 37 °C for 24 h determined by the 3-(4,5-dimethylthiazol-2-yl)-2,5-diphenyltetrazolium bromide (MTT) assay. The data represent the mean ± SD from three replicates.

**Figure 4 molecules-27-03884-f004:**
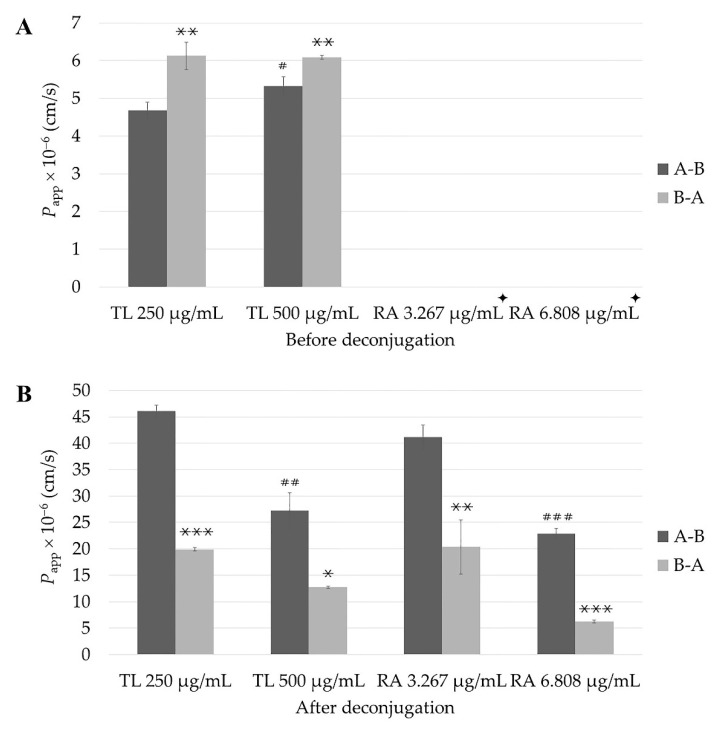
Permeability of RA in 250 and 500 µg/mL TL extracts across Caco-2 cell monolayers in both directions: before deconjugation (**A**) and after deconjugation (**B**). The data represent the mean ± SD from three replicates. *, **, *** were significantly different from A to B direction (*p* < 0.05, 0.01, and 0.001, respectively). ^#^, ^##^, ^###^ were significantly different from RA in 250 µg/mL TL extract or 3.267 µg/mL pure RA (*p* < 0.05, 0.01 and 0.001, respectively). ^✦^ RA could not be detected due to below LLOQ.

**Figure 5 molecules-27-03884-f005:**
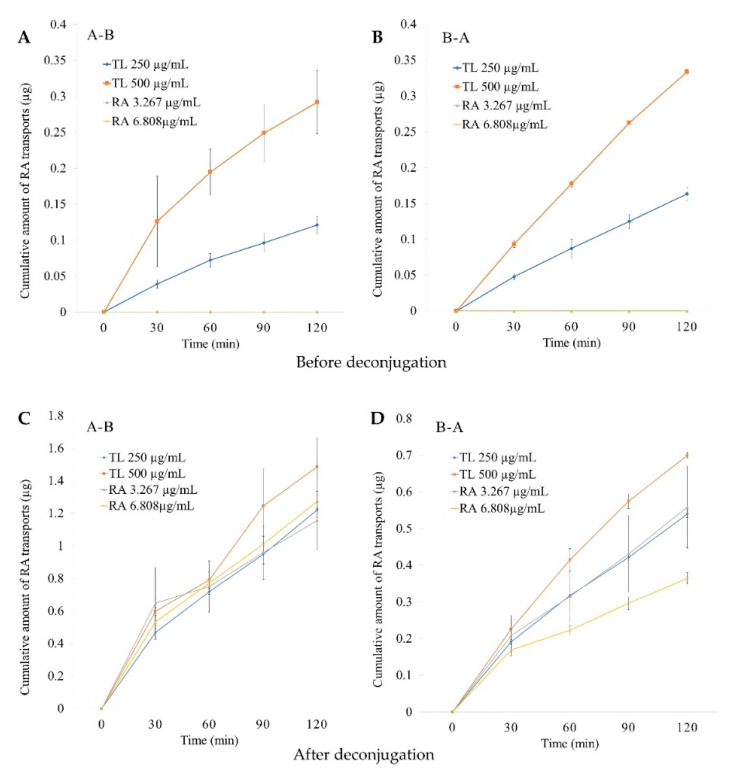
Cumulative amount of RA transports in 250 and 500 µg/mL TL extracts across Caco-2 cell monolayers over time in both directions: A-B (**A**) and B-A (**B**) directions before deconjugation, and A-B (**C**) and B-A (**D**) directions after deconjugation. The data represent the mean ± SD from three replicates.

**Figure 6 molecules-27-03884-f006:**
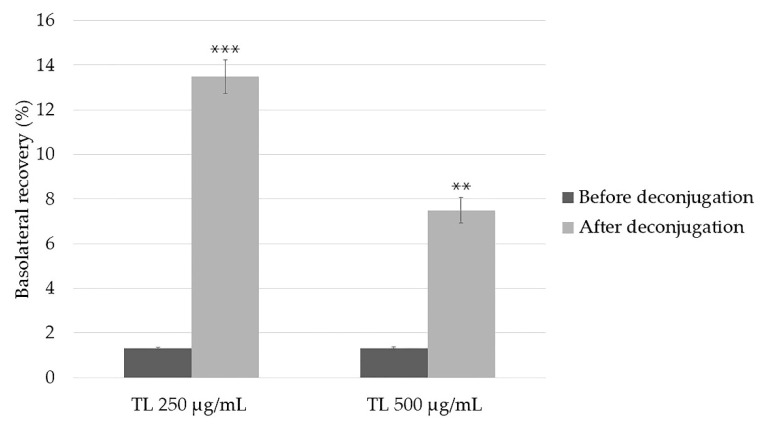
The basolateral recoveries of RA in 250 and 500 µg/mL TL extracts after deconjugation with β-glucuronidase/sulfatase. Data are the mean ± SD from three replicates. ** and *** were significantly different from data obtained for samples, which were not deconjugated (*p* < 0.01 and 0.001, respectively).

**Table 1 molecules-27-03884-t001:** Percentages remaining of RA in TL extracts for short-term (stored at room temperature for 24 h) and long-term (stored at −80 °C for 30 days).

RA Concentration (µg/mL)	RT, 24 h	−80 °C, 30 Days
0.05	99%	97%
0.8	99%	98%

**Table 2 molecules-27-03884-t002:** The *P*_app_ of RA in TL extracts and pure RA before and after deconjugation.

Tests	*P*_app_ × 10^−6^ (cm/s) (before Deconjugation)		ER	*P*_app_ × 10^−6^ (cm/s) (after Deconjugation)		ER
	A-B Direction	B-A Direction		A-B Direction	B-A Direction	
RA in TL extract						
250 µg/mL	4.7 ± 0.2	6.1 ± 0.4 **	1.3	46.1 ± 1.2	19.9 ± 0.3 ***	0.4
500 µg/mL	5.3 ± 0.3 ^#^	6.1 ± 0.1 **	1.1	27.3 ± 3.3 ^##^	12.7 ± 0.2 *^,###^	0.5
RA in pure RA						
3.267 µg/mL	N.A.	N.A.	-	41.2 ± 2.3	20.4 ± 5.2 **	0.5
6.808 µg/mL	N.A.	N.A.	-	22.9 ± 1.0 ^###^	6.2 ± 0.2 ***^,#^	0.3

*P*_app_ values in cm/s for RA quantitated in TL extract and pure RA before and after deconjugation in both A-B and B-A directions. Values represent the mean of three independent replicates ± standard deviation (SD). N.A. indicates *P*_app_ = 0 (non-absorbed) as the other presented as the values were below LLOD. Data represent the mean ± SD from three replicates. *, **, *** were significantly different from A-B direction (*p* < 0.05, 0.01, and 0.001, respectively). ^#, ##, ###^ were significantly different from RA in 250 µg/mL TL extracts or 3.267 µg/mL pure RA (*p* < 0.05, 0.01, and 0.001, respectively).

## Data Availability

Not applicable.

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
