# Peer review of "Evaluation of the Intestinal Permeability of Rosmarinic Acid from Thunbergia laurifolia Leaf Water Extract in a Caco-2 Cell Model"

_molecules, 2022, doi:10.3390/molecules27123884_

Round 1
Reviewer 1 Report
In this paper the authors have investigated the permeability of Thunbergia laurifolia water extracts and rosmarinic acid in a Caco-2 cell monolayer model, both in free and conjugated forms. The study is conducted using common methodology in the field. The manuscript is carefully conceived, clearly divided into sections and the results are well explained. However, there are some points that the authors should address:
- All the abbreviations used have to be fully given when they first appear, both in the Abstract and in the body of the text (e.g. MTT first appears in line 126, Figure 3 and is fully given in Materials and Methods section; AP, BL first appear in line 136 and is fully given in Materials and Methods section
- Line 137 typo mistake 5.33 ± 0.25 x 10-6 x 10-6
- Line 145, Figure 4, the first word is missing
- Line 146 typo mistake – the data
- Figure 5 needs resolution improvements
I would also like to recommend the authors to use as references more recent papers (2020-2022), such as:
https://www.mdpi.com/2076-3921/9/1/47
https://www.mdpi.com/2076-3921/10/11/1678/htm
https://www.mdpi.com/1420-3049/27/10/3292/htm
https://www.sciencedirect.com/science/article/pii/S1319562X20303636
Author Response
Dear Editor and reviewer,
First of all, we would like to thank you for your valuable comments and suggestions on our manuscript. We have addressed the concerns raised by the reviewer by adding explanations to each of the queries. Our response is in a point-by-point format, as shown below. Line numbers refer to the marks made up using the "Track Change" in the red-colored version of the revised manuscript. The updated texts in this manuscript are also using the track version. For the English language and style, we did recheck the spelling and revise/improve minor language styles.

Reviewer 2 Report
Summary:
Authors investigated for the first time the permeability of rosmarinic acid from Thunbergia laurifolia, a plant which is poorly investigated and often used as folk medicine in Thailand and India. The paper brings the important findings and is worth publishing. However, the paper could and should be improved in order to make the findings understandable. Therefore, I suggest reconsidering the article after Major Revisions that are mentioned in detail together with Minor Revisions which are not underlined.
Comments
Title: I suggest modifying the title in order to describe the content of the article more accurately as follows: Evaluation of the intestinal permeability of rosmarinic acid from Thunbergia laurifolia leaves water extract in Caco-2 cell model
Abstract: Abstract is misleading in some parts. It should reflect more specifically on the findings in the article and give more accurately the description of the deconjugation procedure.
Line 21: Point out the reason why RA was quantified in TL extract.
Line 23: From the sentence one could conclude that not only RA was followed in this work but also other components in TL extract. Rephrase the sentence to indicate that only permeability of RA was investigated IN TL extract and as pure compound. Change through the text to distinguish between different sources of RA.
Line 24: The sentence is misleading. I suggest keeping only the part with detection which also can be connected to the previous sentence.
Line 26: The sentence is misleading. I would point out what apparently happened with RA in Caco-2 cells and how you managed to confirm that.
Introduction:
Line 46: Previous studies ‘SHOWED THAT’ …
Line 47-49: Separate the groups of analytes with ‘;’ between different extracts.
Line 51: Put ‘RA’ instead ‘It’.
Line 53: Remove ‘and’.
Line 54: Remove ‘with interesting biological activities’ and instead put a sentence which describe the applications of RA in practice.
Line 55: Put ‘such’ instead of ‘e.g.’.
Line 54-57: I would suggest rephrasing to ‘It is characterized with broad pharmacological activities what makes it interesting in drug development.’.
Line 58-82: I would suggest completely reorganization of this part and inserting the answers to these questions: 1. What is known about the BA of RA (high-low; what can impact it in general and from plant extract matrix), 2. Why is the absorption important to BA and why Caco-2 cell model is used (also mention the limitations and point out the enzymes that are expressed in the model that are important for metabolism).
Line 60: Replace ‘clear’ with ‘investigated’.
Line 62: Insert ‘in this study’ before ‘to investigate’.
Line 63: RA extracts?
Line 64: Insert ‘,’ after standard.
Line 72: Briefly describe the BA in previous studies.
Line 73: Rephrase ‘types and amounts of composition’
Line 85: There is no ‘above’ HPLC conditions.
Line 86: What signal was followed? What is compared in the calibration curve?
Line 98-100: Mention also in the materials.
Line 98-114: I suggest merging these two paragraphs and make a distinction between stability over time and the stability in HBSS.
Line 109-110: Mention also in the materials.
Line 111: ‘FROM the extract and AS pure’.
Table 1: Remove decimal places for values which are presented as percentage and higher than 10 %. For values lover than 10%, leave only one decimal place. Change throughout the text.
Line 111: Also put the absolute values.
Figure 2. * put directly over the standard deviation mark. Change throughout the figures. Change explanation to ‘After 2h-incubation’. Description: ‘The data REPRESENT’.
Line 122: Put % and remove decimal places. Point out which concentrations were selected for permeability assay.
Figure 3. Is the y-axis correct for RA concentrations? Remove decimal places on y-axis. Description: ‘…RA was investigated on Caco-2 cells after incubation at 37 °C for 24 h determined by the MTT assay.’
Line 127: I would suggest changing the title of the paragraph to describe more correctly its meritum: ‘The impact of post permeability deconjugation treatment with glucuronidase/sulfatase on Papp of RA from TL extract and of pure compound’.
Line 130: Change sentence to ‘TEER values gradually increased over time and reached 400 Ω*cm2 in 21 days.’.
Line 133: In vitro permeability study
Line 135: Rephrase to: ‘The Papp values of RA from TL extract in concentration of 250 µg/mL…’
Line 136: space missing after 4.68
Line 137: ‘…, while in TL extract in concentration of 500 µg/mL …’. Remove duplicate *10^-6
Line 138-143: The explanation is not clear.
Figure 4. First word missing. Line 145: ‘IN both directions’. Line 146: ‘The DATA’.
Line 157 and 159: ‘… RA as pure compound in concentration of..’
Line 159: put ‘Papp’ instead of ‘values’.
Line 160: space before 2 is missing; erase ‘their’.
Line 161: IN concentrationS
Line 162: 2.02-
Line 164-171. Describe how cumulative amounts were calculated.
Line 180: Reprase the sentence to: ‘. Deconjugation resulted in about 10 and 5 times….’
Figure 6. Description: ‘…after DECONJUGATION WITH…’, erase ‘over 2 h incubation’, replace ‘before the enzyme treatment’ to ‘data obtained for samples which were not deconjugated’
Line 189: IN concentrationS
Line 190: ‘…RA as pure compound in 3.267 µg/mL and 6.808 µg/mL were...’
Line 193: ‘…after deconjugation.’
Table 2. Space missing after 4.68. Line 198: ‘...as the values were below LOD’. Is font of # correct?
Discussion: Parts should be changed according to the changes in the Results section. The findings should be explained more precisely.
Line 204: ‘In this study we demonstrated for the first time the intestinal permeability of RA from TL leaf water extracts and pure RA by using in vitro Caco-2 cell model’ (Also decide to keep either TL ‘leaf’ or ‘leaves’. Change through the text.)
Line 206-209: Make the main conclusions directly from the Results.
Line 211-213: Data not presented in the Results.
Line 220: The sentence is not clear.
Line 227: Then how is it different from the absorption of the pure RA?
Line 237: The sentence is not clear.
Line 240: Why is this important to RA? Refer to the introduction and explanation of RA BA?
Line 242: What about paracellular transport?
Line 244-247: This part should be mentioned in the introduction.
Line 244-256: I would not agree with these explanations. I think it is important to distinguish between low absorption and intensive 1. Phase metabolism. The interesting conclusion could be that the investigation showed that one could conclude that the molecule has low permeability, but the real situation includes good permeability but also intensive conjugation in Caco-2 cells. Therefore, this investigation is stresses out the importance of deconjugation step in permeability assays for molecules prone to conjugation.
Materials: All in all, well written with couple of minor changes to be made.
Line 280: Put city and state for all the manufacturers mentioned in the work.
Line 282: HSSS is miswritten?
Line 287: MILLPORE is miswritten?
Line 291: Were the samples dissolved in HBSS or culture media?
Line 299: Was the temperature ‘room temperature’ or 25°C?
Line 312: Caco-2 misspelled.
Line 316: …(MTT) following the modified assay
Line 318: Mention the maximum DMSO concentration in culture media.
Line 320: Is this concentration of TL extract or pure RA?
Line 323: How much of MTT solution was put and in what concentration?
Equation (1): Instead of signal and background place the actual value which was measured.
Line 339: Mention the minimal TEER value that was measured a day after the experiment.
Line 365: 37 °C
Line 388: Explain the abbreviation.
Conclusions: This chapter should be completely rewritten according to the Discussion.
References: Bold the year in Lines 452, 477.
Author Response
Dear Editor and Reviewer,
First of all, we would like to thank you for your valuable comments and suggestions on our manuscript. We have addressed the concerns raised by the reviewer by adding explanations to each of the queries. Our response is in a point-by-point format, as shown below. Line numbers refer to the marks made up using the "Track Change" in the red-colored version of the revised manuscript. The updated texts in this manuscript are also using the track version. For the English language and style, we did recheck the spelling and revise/improve minor language styles.

Round 2
Reviewer 2 Report
The paper could and should be improved in order to make the findings understandable.
Major revision |
Introduction part
Line 62-96: The answers to the question are still not well addressed: 1. What is known about the BA of RA. It would be useful to mention other in vitro studies in which RA BA was investigated in pure form or in other plants.
Result part
Line 156: ‘Significant differences’ should be ‘significantly different’
Disscussion part
Line 237-242: Sentences are not clear. English should be improved and the information should not be repeated.
Line 242-245: The sentence is still not clear.
Line 256-257: The sentence is not clear.
Line 262-268: The sentences are still not clear.
Line 278-281: The sentence is not clear.
Line 281-286: The information given in this paragraph are contradictory to the one given in the introduction part.
Line 289: ‘less’ should be ‘low’?
Line 293-295: The sentence is not clear.
Conclusion part
Line 438-451: RA penetrated through Caco-2 cell monolayers AND WAS detected by HPLC-PDA. RA in TL leaf water extracts could be detected in both A-B and B-A direction UNLIKE pure RA. Thus, this study IS EVIDENCE OF the extensive biotransformation of RA while penetrating through Caco-2 cells.
Minor revision |
Line 23: Erase ONLY.
Line 44: …for treating intoxications with insecticides, herbicides, lead, alcohol, as well as chemical toxins.
Line 58: ..such AS..
Comment number 28: This comment should be: “Figure 3. Is the X-axis correct for RA concentrations˝. Are you sure that the cytotoxicity of both pure RA and several times less concentration in TL extract is the same?
Comment number 57: “Line 320: Is this concentration of TL extract or pure RA?”
Comment number 59: “Equation (1): Place the actual value of the signal (absorbance) which was measured.” I.e. A (sample) – A (blank)…
Author Response
Dear Editor and Reviewer 2,
Thank you so much for your valuable comments and suggestions on our work. We appreciate it so much. Please see the attached files for a point-by point response to the reviewer’s comments.
Best regards,
Korbtham Sathirakul
